# Efficacy and safety of non-pharmacological therapy under the guidance of TCM theory in the treatment of anxiety in patients with myocardial infarction: A protocol for systematic review and meta-analysis

**Weizhe Zhao** [1,2], **Yue Wang** [1], **Jiqiu Hou** [2], **Wanli Ding** [1], **Wendong Suo** [1], **Zhu Liu** [1], **Yutong Zhou** [3], **Haibin Zhao** [2]*

1 Beijing University of Traditional Chinese Medicine, Beijing, China, 2 Dongfang Hospital of Beijing University of Traditional Chinese Medicine, Beijing, China, 3 Guang'anmen Hospital, Chinese Academy of Traditional Chinese Medicine, Beijing, China

☯ These authors contributed equally to this work.

* zhaoweizhe@126.com

**Data Availability Statement:** No datasets were generated or analysed during the current study. All

## Abstract

### Background

With the increasing pressures of modern life and work, combined with a growing older population, the incidence of comorbid anxiety and myocardial infarction (MI) is increasing. Anxiety increases the risk of adverse cardiovascular events in patients with MI and significantly affects their quality of life. However, there is an ongoing controversy regarding the pharmacological treatment of anxiety in patients with MI. The concomitant use of commonly prescribed selective serotonin reuptake inhibitors (SSRIs) and antiplatelet medications such as aspirin and clopidogrel may increase the risk of bleeding. Conventional exercise-based rehabilitation therapies have shown limited success in alleviating anxiety symptoms. Fortunately, non-pharmacological therapies based on traditional Chinese medicine (TCM) theory, such as acupuncture, massage, and qigong, have demonstrated promising efficacy in treating MI and comorbid anxiety. These therapies have been widely used in community and tertiary hospital settings in China to provide new treatment options for patients with anxiety and MI. However, current studies on non-pharmacological TCM-based therapies have predominantly featured small sample sizes. This study aims to comprehensively analyze and explore the effectiveness and safety of these therapies in treating anxiety in patients with MI.

### Method

We will systematically search six English and four Chinese databases by employing a predefined search strategy and adhering to the unique rules and regulations of each database to identify studies that fulfilled our inclusion criteria, to qualify for inclusion, patients must be diagnosed with both MI and anxiety, and they must have undergone non-pharmacological TCM therapies, such as acupuncture, massage, or qigong, whereas the control group

relevant data from this study will be made available upon study completion.

**Funding:** The author(s) received no specific funding for this work

**Competing interests:** The authors have declared that no competing interests exist.

received standard treatments. The primary outcome measure will be alterations in anxiety scores, as assessed using anxiety scales, with secondary outcomes encompassing the evaluations of cardiopulmonary function and quality of life. We will utilize RevMan 5.3 to conduct a meta-analysis of the collected data, and subgroup analyses will be executed based on distinct types of non-pharmacological TCM therapies and outcome measures

## Results

A narrative summary and quantitative analysis of the existing evidence on the treatment of anxiety patients with MI using non-pharmacological therapies guided by Traditional Chinese Medicine theory.

## Conclusion

This systematic review will investigate whether non-pharmacological interventions guided by TCM theory are effective and safe for anxiety in patients with MI, and provide evidence-based support for their clinical application.

## Systematic review registration

PROSPERO CRD42022378391

## Introduction

According to an epidemiological report released by the World Health Organization (WHO) in December 2020, ischemic heart disease remains the leading cause of death globally, accounting for 16% of deaths worldwide [1]. In China, there were 1.7869 million reported cases of myocardial infarction(MI) in 2020, according to the "2020 China Cardiovascular Disease Report" issued by the National Health Commission [2]. Anxiety, a common psychological disorder characterized by episodic or continuous tension, fear, and worry, is prevalent in patients with cardiovascular disease [3]. Research has demonstrated that the likelihood of experiencing depression and anxiety after MI is approximately three times greater than in the general population [4]. About 26% of out-of-hospital cardiac arrest (OHCA) survivors exhibit anxiety [5]. Furthermore, over half of the patients display anxiety symptoms within one week following an AMI [6]. Moreover, anxiety in patients with MI is closely associated with poor cardiac prognosis and higher mortality rates [7], and patients who experience anxiety symptoms after MI have an increased risk of new cardiovascular events or death by as high as 36% [8]. Furthermore, negative emotional expressions such as anxiety can significantly reduce patients' quality of life and long-term psychological adjustment ability and increase medical expenses [4]. Therefore, the effective control of anxiety associated with MI has become a pressing issue in clinical practice.

In 2008, the American Heart Association (AHA) issued recommendations for screening all patients with coronary heart disease for anxiety and depression, while the European Clinical Practice Guidelines explicitly suggested treating depression and anxiety in patients with heart disease [9]. Currently, the commonly used clinical treatment methods mainly include drug therapy, which aims to restore nervous system dysfunction through oral or injectable administration of chemical drugs, and psychotherapy, which involves establishing psychological connections with patients and guiding them to understand and cope with negative emotions.

However, studies have revealed that selective serotonin reuptake inhibitors (SSRIs), a representative class of psychotropic medications, such as sertraline, fluoxetine, and fluvoxamine, pose an increased risk of bleeding when co-administered with the most frequently used anti-platelet agents after a heart attack, including aspirin or clopidogrel, which can cause bleeding events such as gastrointestinal bleeding and hemorrhagic stroke [10]. Furthermore, several psychotropic drugs, such as benzodiazepines and anticonvulsants, have various limitations, including withdrawal effects, numerous side effects, poor medication adherence, and debatable effects on the prognosis of heart disease [11]. Consequently, many patients prefer alternative therapies with fewer side effects and greater safety [12, 13]. However, studies have revealed that current psychological therapies have limited efficacy in patients with heart disease and comorbid anxiety [14]. Moreover, some exercise therapies, resembling cardiac rehabilitation, have demonstrated effectiveness comparable to placebos in the treatment of anxiety [15]. Consequently, an increasing number of individuals are turning to non-pharmacological therapies guided by Traditional Chinese Medicine (TCM) theories.

TCM explains the intricate relationship between MI and emotional disorders from the perspectives of Qi, blood, and meridians. TCM theory posits that the unhindered circulation of Qi and blood along the meridians is vital for life activities, and pathological conditions like "Blood stasis" and "Qi stagnation" that block the meridians can induce a range of physical and psychological ailments [16]. According to TCM theory, "Blood stasis" is the primary pathological product of MI, obstructing the heart meridian and resulting in symptoms of chest pain and tightness, "blood stasis" can also affect the normal flow of Qi in the meridians, leading to abnormal emotions such as anxiety [17]. To address this complex situation, TCM employs non-pharmacological therapies that differ from herbal treatments. Evidence has suggested that these therapies can enhance psychological and emotional well-being by "regulating Qi and blood circulation in the meridians" while minimizing adverse reactions [18–20].

Non-pharmacological therapies in TCM are external treatment methods based on the traditional Chinese philosophies of Yin-Yang and the Five Elements and, TCM's unique concepts of holistic therapy and pattern differentiation-based treatment. These therapies mainly involve manipulative and movement-based interventions. Manipulative interventions include acupuncture, massage, cupping, and guasha (scraping therapy), whereas movement-based interventions include qigong, tai chi, and the Five Animal frolics, etc. [21, 22]. These therapies possess inherent TCM characteristics, such as emphasizing systemic holistic treatment, individualized treatment based on pattern differentiation, and dynamic treatment that considers a patient's status and changes [23].

In China, non-pharmacological therapies guided by TCM theory are widely used across various levels of healthcare institutions, from community health centers to tertiary hospitals. They are highly accepted by patients [24], and people are willing to embrace modalities such as acupuncture, massage, and tai chi as treatments for anxiety and depression [25]. Regarding efficacy, some clinical trials have demonstrated the effect of TCM non-pharmacological therapies in treating post-MI anxiety; however, most studies have been conducted with small sample sizes at single centers, and high-quality studies with large sample sizes are still lacking. Therefore, we conducted a meta-analysis to evaluate the efficacy and safety of TCM non-pharmacological therapies for the treatment of anxiety in patients with MI.

## Methods

### Study registration

The protocol for this systematic review and meta-analysis was drafted following the preferred reporting items for systematic reviews and meta-analyses and was registered with the

international platform for systematic review and meta-analysis protocols (Registration Number: CRD42022378391) on 4 December 2022.

## Qualification criteria

**Research design.**   Randomized Controlled Trial (RCT). The study will be conducted using English and Chinese languages only.

**Study participants.**   Inclusion criteria were as follows: (1) Patients clinically diagnosed with myocardial infarction, including acute coronary syndrome, who have undergone percutaneous coronary intervention or coronary artery bypass graft surgery for coronary heart disease. (2) Patients with a clinical compliance anxiety scale score of anxiety level (such as HADS-A> 7) and/or a DSM-5 primary diagnosis of anxiety disorder were eligible (participants completed DSM-5 structured clinical interviews and anxiety assessment scales such as the 14-item Hamilton Anxiety Rating Scale). (3) The intervention and control groups should have a good balance in terms of sex, age, disease course, and basic medical conditions, with comparable data.

**Exclusion criteria.**   (1) Patients with severe heart failure who are unable to receive the research intervention. (2) History of mania, consciousness disorders, or other serious mental illnesses including schizophrenia, bipolar disorder, or severe major depressive disorder. Patients with consciousness disorders may be unable to communicate accurately during scale evaluations, and other severe mental illnesses could potentially influence their responses to treatment or interfere with anxiety symptoms, thereby affecting research outcomes. Furthermore, the efficacy of TCM on this patient population remains unclear. Therefore, excluding these patients helps to mitigate the risk of exacerbating their condition. (3) Literature that cannot provide detailed outcome indicators. (4) Duplicate publications.

## Intervention measures

**Intervention group.**   Patients will receive non-pharmacological traditional Chinese medicine therapies, including acupuncture, acupoint application, massage, traditional Chinese exercise therapy (such as tai chi, "eight-section brocade", and "five animal frolics "), and other methods, either alone or in combination (such as massage combined with acupuncture).

**Control group.**   Patients will receive general treatments, observation, education, general exercise, sham acupuncture (a commonly used control group in acupuncture clinical trials), and other treatment methods.

**Outcome indicators.**   The main outcome indicator will be anxiety levels, assessed using anxiety scales (HADS-A, HAMA, SAS, GAD-7, and DASS-A). Secondary outcome indicators include cardiopulmonary function (such as 6-minute walk test and peak oxygen consumption), quality of life (assessed using the 36-item Short Form Health Survey or any other valuable scales), and the incidence of adverse events, such as liver or kidney damage, nausea, vomiting, and constipation.

## Search strategy

Our study will adhere to the Cochrane Handbook and the Preferred Reporting Items for Systematic Reviews and Meta-analysis Protocols (PRISMA-P) guidelines, with two independent evaluators systematically searching for articles published in PubMed, Embase, Ovid, Scopus, Cochrane Central Register of Controlled Trials, Web of Science, China Integrated Knowledge Resources Database, Chinese scientific journals database, Chinese biomedical literature database, and Wanfang Database from their inception until December 31, 2022.The primary search terms will encompass "myocardial infarction", "anxiety", "anxiety state", "generalized

anxiety disorder", "GAD", "traditional Chinese medicine", "non-pharmacological therapy", "acupuncture", "moxibustion", "massage", "qigong", "tai chi", "eight-section brocade", and "five animal frolics" [26–28]. The search strategy for PubMed is detailed in Table 1. Similar terms will be translated into Chinese to cater to Chinese databases. The language utilized in this review will be restricted to English and Chinese.

## Study selection

Two independent reviewers will screen the identified articles based on the eligibility criteria by assessing the titles and abstracts. Further evaluation of potentially eligible articles will be conducted through full-text review. The process of study selection will be illustrated in Fig 1. Disagreements will be resolved through discussion between the reviewers.

## Data extraction

The included studies will be independently assessed by two reviewers for the following data: reference information (first author, publication year, etc.), study characteristics (study objectives, randomization methods, blinding, etc.), participant characteristics (sufficiently described demographics, age, sex, classification of coronary heart disease, etc.), intervention details (type of traditional Chinese medicine, non-pharmacological treatments, control intervention, intervention time and frequency, follow-up time, etc.), outcome measures, and adverse events. Any

**Table 1. Search strategy for PubMed.**

| NO. | Search Terms |
|---|---|
| #1 | #1 Miocardial infarction [mh] OR Miocardial infarction [tiab] OR Disease*, Infarction, Myocardial [tiab] OR Infarction, Myocardial[tiab] OR Myocardial Infarctions[tiab] OR Cardiovascular Stroke[tiab] OR Cardiovascular Strokes[tiab] OR Stroke, Cardiovascular[tiab] OR Myocardial Infarct[tiab] OR Infarct, Myocardial[tiab] OR Angina, Unstable[mh] OR Anginas, Unstable[tiab] OR Unstable Anginas[tiab] OR Angina Pectoris, Unstable[tiab] OR Angina Pectori, Unstable[tiab] OR Unstable Angina Pectori[tiab] OR Unstable Angina[tiab] OR Angina at Rest[tiab] OR Angina, Preinfarction[tiab] OR Preinfarction Angina [tiab] OR Preinfarction Anginas[tiab] OR Myocardial Preinfarction Syndrome[tiab] OR Myocardial Preinfarction Syndromes[tiab] OR Preinfarction Syndrome, Myocardial[tiab] OR Syndrome, Myocardial Preinfarction[tiab] OR Chronic Stable Angina[tiab] OR Angina Pectoris, Stable[tiab] OR Stable angina[tiab] OR Syndromes, Myocardial Preinfarction[tiab] OR percutaneous coronary intervention[tiab] OR PCI[tiab] OR coronary artery bypass grafting[tiab] |
| #2 | Anxiety[mh] OR Anxiety[tiab] OR Social Anxiet*[tiab] OR Anxiet*, Social[tiab] OR Hypervigilance[tiab] OR Nervousness[tiab] OR Anxiousness[tiab] OR Performance Anxiety[mh] OR Anxieties, Performance[tiab] OR Anxiety, Performance[tiab] OR Performance Anxieties[tiab] OR panic [tiab] OR obsessive compulsive disorder [tiab] OR generalized anxiety disorder [tiab] OR GAD[tiab] OR agoraphobia[tiab] OR phobic disorders[tiab] OR stress disorders[tiab] OR post-traumatic stress disorder[tiab] OR PTSD[tiab] |
| #3 | #1 AND #2 |
| #4 | Complementary Therapies [mh] OR Complementary [tiab] OR Therapies, Complementary [tiab] herapy, Complementary [tiab] OR Alternative Medicine [tiab] OR Therapies, Alternative[tiab] OR Therapy, Alternative [tiab] OR Non-pharmacological therapy [mh] OR Nonpharmacological therapy [mh] OR Non-pharmacological therapy [tiab] OR Nonpharmacological therapy [tiab] |
| #5 | Acupuncture [mh] OR Acupuncture* [tiab] OR Acupuncture [tiab] OR Massage [mh] OR Massage* [tiab] OR massage [tiab] OR acupoint [tiab] OR Traditional Chinese Medicine Utilitarian [tiab] OR utilitarian [tiab] OR Taiji [mh] OR Taiji[tiab] OR "Tai Chi"[tiab] OR "Tai-Ji"[tiab] OR "Tai-Chi"[tiab] OR Qigong [mh] OR Qigong[tiab] OR Wuqinxi [tiab] OR Tendon Changing Classic [tiab] OR Yijinjing[tiab] |
| #6 | #4 OR #5 |
| #7 | Randomized controlled trial[pt] OR controlled clinical trial[pt] OR randomized[tiab] OR placebo[tiab] OR randomly[tiab] OR trial[tiab] OR groups[tiab] |
| #8 | meta analysis[tiab] OR cochrane review[tiab] OR systematic review[tiab] |
| #9 | #7 OR #8 |
| #10 | #3 AND #6 AND #9 |

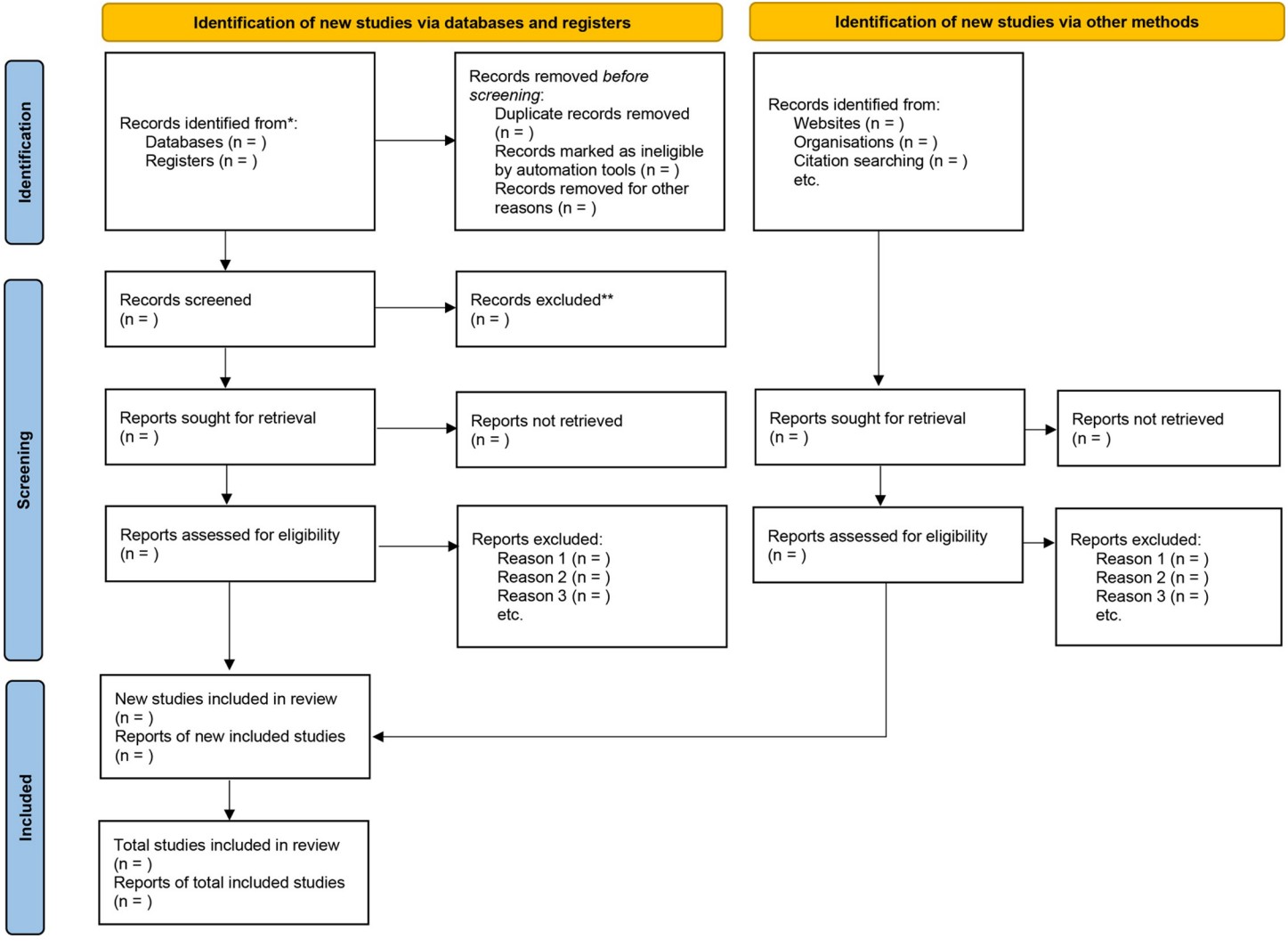

**Fig 1. PRISMA flowchart of studies selection.**

discrepancies will be resolved through discussion between the reviewers. If there is disagreement between the two reviewers, a third reviewer will be consulted.

## Dealing with missing data

Two reviewers will contact the corresponding authors by email or phone to obtain missing or uncertain data. If the data cannot be obtained, the study will be excluded. The potential impact of missing data on the overall analysis will be assessed using sensitivity analysis.

## Risk of bias assessment

The quality of eligible studies will be assessed independently by two reviewers using the Cochrane Risk of Bias tool for RCTs. Mainly five domains will be assessed: selection, performance, detection, attrition, reporting, and other biases. The risk of bias will be classified as "low", "high", or "unclear". Discrepancies will be resolved through discussion.

## Assessment of publication bias

If at least 10 studies are available for meta-analysis, potential publication bias will be analyzed and displayed using a funnel plot. Egger's test will also be used to assess publication bias, with a P value of less than 0.05 indicating significant bias

## Data analysis

Data analysis will be carried out using RevMan software (version 5.3) provided by the Cochrane Collaboration Network, specifically designed for meta-analyses. Binary outcome indicators will be represented as relative risk (RR), while continuous outcome indicators will employ the mean difference (MD) for the effect size when all studies use the same measurement units. However, suppose various methods, such as different psychiatric scales (HADS-A, HAMA, SAS, GAD-7, and DASS-A), are used to measure the same outcome. In that case, the standardized mean difference (SMD) will be utilized. SMD allows for variability between different scales by standardizing the results, thereby enabling the comparison and combination of outcomes from various instruments.

The heterogeneity of the included studies will be assessed using the $I^2$ statistic and Cochrane Q test. Suppose the $I^2$ value is below 50%, and the P-value is above 0.1. This will be interpreted as the absence of statistical heterogeneity, and a fixed-effects model will be used for meta-analyses. Conversely, if the $I^2$ value is above 50%, indicating statistical heterogeneity, subgroup or sensitivity analyses will be conducted to identify and address the potential heterogeneity factors. If statistical heterogeneity persists, but clinical homogeneity is maintained, a random-effects model will be adopted for the meta-analyses [29].

In the event of high heterogeneity during the meta-analysis process, several steps will be taken: 1) We will perform a subgroup analysis, categorizing studies based on potential sources of heterogeneity, followed by separate meta-analyses for each subgroup. 2) Meta-regression methods will be used to analyze study characteristics and identify potential features contributing to heterogeneity. 3) If the issue of high heterogeneity cannot be resolved effectively, we will consider converting the meta-analysis into a systematic review, thereby offering a qualitative synthesis of studies without quantitative aggregation.

**Subgroup analysis.** Subgroup analysis will be performed based on different types of traditional Chinese nonpharmacological treatments (e.g., acupuncture treatment, acupoint application, massage Tuina, and utilitarian exercise group). We will also conduct subgroup analyses based on the severity of anxiety symptoms in the included patients and the psychiatric scales used in the research. Patients with myocardial infarction and comorbid anxiety will be categorized into younger (<40 years) and older (≥40 years) age groups.

**Sensitivity analysis.** Sensitivity analysis will be performed to assess the reliability of the synthesized results. We will sequentially exclude low-quality studies and perform an effect size synthesis and meta-analysis again to observe whether there is a significant change in the results, thereby evaluating the stability of the findings.

**Grading the quality of evidence.** In this study, the Cochrane Collaboration's recommended tool for assessing bias risk will be utilized to grade the quality of evidence. Two independent reviewers will evaluate the literature quality based on seven aspects: random sequence generation, allocation concealment, blinding of participants and personnel, blinding of outcome assessment, data integrity, selective reporting, and other biases (such as conflicts of interest and funding sources). The quality of evidence will be classified as low risk, high risk, or unclear risk based on the evaluation results. Cross-checking will be conducted after completion, and ambiguities will be discussed between reviewers or referred to a third authoritative expert for consultation.

**Ethics and dissemination.** This study is a systematic review, which generally does not require ethical approval. Moreover, the study will not involve any private data or interventions on the participants. The final results of the analysis will be disseminated through the PROS-PERO website and peer-reviewed journals.

## Patient and public involvement

The design of this study does not include the involvement or representation of patients or the public.

## Amendments

This report will involve the search and selection of studies, extraction and management of data, and analysis of the data. If there are any changes to the original plan of the systematic review during the process, they will be promptly updated on the PROSPERO website and formally stated in the final report with the reasons and date of the change.

## Discussion

With the increasing pressure of work and the acceleration of an aging society, the incidence of MI combined with anxiety has risen due to multiple factors such as high psychological stress, high-fat diet, and disrupted circadian rhythms. Anxiety is an independent risk factor for MI and an important prognostic factor. Although the pathological mechanism of anxiety in patients is unclear, domestic and foreign scholars have proposed the importance of screening for and treating anxiety and depression in patients with MI. Non-pharmacological TCM therapies are widely used in China and have shown excellent clinical therapeutic effects. However, most related studies are single-center, small-sample studies, leaving room for further evidence-based medical research. This systematic review will strictly screen and comprehensively analyze eligible RCTs to evaluate the safety and effectiveness of non-pharmacological TCM therapies for treating anxiety in patients with MI. Furthermore, sub-group analyses will be conducted to determine whether these therapies can provide additional treatment options for clinical practitioners.

Nonetheless, this study also presents some potential limitations: We have only included literature in Chinese and English, which may exclude relevant studies in other languages that potentially use non-pharmacological TCM therapies. Additionally, our study exclusively assesses the outcomes of RCTs without incorporating results from other types of studies such as cohort studies and observational trials. Our study aims to lay the groundwork for more in-depth research in the future, providing new evidence-based medical guidelines for clinical applications. Furthermore, it would be beneficial to expand the scope of research by including more languages and different study types, as well as more high-quality research in the future [30], thoroughly considering limitations and biases, and conducting more profound and rational investigations and discussions.

## Supporting information

**S1 File. Search strategies.**
(DOC)

**S1 Table. PRISMA-P 2015 checklist.**
(DOC)

## Author Contributions

**Conceptualization:** Weizhe Zhao, Yue Wang.

**Data curation:** Weizhe Zhao, Yue Wang, Jiqiu Hou, Wendong Suo, Zhu Liu, Yutong Zhou.

**Formal analysis:** Weizhe Zhao, Yue Wang, Yutong Zhou.

**Funding acquisition:** Wendong Suo.

**Investigation:** Weizhe Zhao, Yue Wang.

**Methodology:** Weizhe Zhao, Wanli Ding.

**Software:** Wanli Ding, Haibin Zhao.

**Supervision:** Yue Wang, Wendong Suo, Zhu Liu.

**Validation:** Weizhe Zhao, Yue Wang.

**Visualization:** Weizhe Zhao, Yue Wang, Haibin Zhao.

**Writing – original draft:** Weizhe Zhao, Yue Wang, Wanli Ding, Zhu Liu, Yutong Zhou.

**Writing – review & editing:** Weizhe Zhao, Yue Wang, Wendong Suo.

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
