## [Decision Letter · Decision Letter 0]

10 Feb 2023

PONE-D-22-34677

Efficacy and safety of non-pharmacological therapy under the guidance of TCM theory in the treatment of anxiety in patients with myocardial infarction: a protocol for systematic review and meta-analysis

PLOS ONE

Dear Dr. Zhao,

Thank you for submitting your manuscript to PLOS ONE. After careful consideration, we feel that it has merit but does not fully meet PLOS ONE’s publication criteria as it currently stands. Therefore, we invite you to submit a revised version of the manuscript that addresses the points raised during the review process.

We look forward to receiving your revised manuscript.

Kind regards,

Qin Xiang Ng, MD, MPH

Academic Editor

PLOS ONE

Journal Requirements:

Additional Editor Comments:

Although the protocol paper is topical, you will see that the reviewers have raised several points that would have to be addressed before publication can be advised.

In addition, please note that:

1. The manuscript requires extensive edits for language, grammar and writing style.

2. "... biggest killer worldwide" - please use more formal and scientific language.

3. The authors seem to use CHD and myocardial infarction interchangeably, please know that the two are not the same thing, one is a chronic disease and the other is an acute event.

4. In the introduction, you should provide some reference to previous works and statistics on the prevalence of anxiety after cardiac arrest (citation: pubmed.ncbi.nlm.nih.gov/34826580).

5. Were search terms in Mandarin used for the Chinese databases? Please specify.

6. How would the studies using different psychiatric scales e.g. HADS-A, HAMA, SAS, GAD-7, and DASS-A, be pooled?

7. "Concomitant with the rising social work pressure in people and the accelerated formation of an aging society" - I am not sure what is this supposed to mean. Please rephrase.

8. What exactly is "TCM theory"? This is mentioned a few times but authors did not go into any detail about TCM theory. 

Reviewers' comments:

Reviewer's Responses to Questions

**Comments to the Author**

1. Does the manuscript provide a valid rationale for the proposed study, with clearly identified and justified research questions?

Reviewer #1: Partly

Reviewer #2: Yes

2. Is the protocol technically sound and planned in a manner that will lead to a meaningful outcome and allow testing the stated hypotheses?

Reviewer #1: Partly

Reviewer #2: Partly

3. Is the methodology feasible and described in sufficient detail to allow the work to be replicable?

Reviewer #1: Yes

Reviewer #2: No

4. Have the authors described where all data underlying the findings will be made available when the study is complete?

Reviewer #1: No

Reviewer #2: Yes

5. Is the manuscript presented in an intelligible fashion and written in standard English?

Reviewer #1: No

Reviewer #2: Yes

6. Review Comments to the Author

You may also provide optional suggestions and comments to authors that they might find helpful in planning their study.

Reviewer #1: 

The title is slightly confusing and it is unclear if it is related to non pharmacological therapies or TCM.

It will benefit from amendment to reflect non-pharmacological TCM usage.

Abstract

- It may be better to change increase in people's life to "increase in people's life expectancy"

- The background section is confusing as it makes its less clear if the authors are inspecting patients with myocardial infarction or CHD.

- A brief sentence on why drug therapy for myocardial infarction and anxiety is controversial.

- The authors may wish to elaborate if they are focusing on anxiety symptoms or anxiety disorders or BOTH.

- "Utilitarians" may be better worded as "practitioners"

- The methods does not describe any inclusion or exclusion criteria for this study in particular the PICO.

Introduction

- I think it is important to discern if the authors are looking at as they are managed and treated differently.

- Likewise, discerning whether you are evaluating coronary heart disease patients / patients with recent myocardial infarction is important

-> This is due to the title only reflecting patients with myocardial infarction

- "In China, a considerable number of people have a certain resistance to psychotropic drugs, and they are more willing to choose some treatment methods with fewer side

effects and higher safety" requires a citation

- The authors describe increased bleeding risk with SSRIs. However, what is the risk and has any recommendation been made by guidelines that SSRI should not be used?

- Clarification needs to be made as to which drugs are controversial to heart disease and its implications

- The authors write initially that the effect of psychotherapy on patients with anxiety and MI is unclear. They latter quote it is not satisfactory.

-> There appears to be some contradictions here.

- Some definition needs to be made with regards to what TCM theory means and what non-pharmacological therapy the authors are looking at?

- The last paragraph in the introduction lacks citations

- Given the heterogeneity of data collected, I am uncertain if a meta-analysis may be most appropriate for TCM trials

-> Perhaps a scoping review may be more appropriate.

Methods

- Was the PRISMA 2020 guidelines used?

- For the anxiety, were a psychiatrist involved in the diagnosis?

-> Noted that the authors mentioned a cut off of >=7 on HADS-A for diagnosis for anxiety

-> However, HADS-A is not used for diagnosis of anxiety disorders and instead valiated for diagnosis of anxity symptoms

- What are the exclusion for participants in the study?

- What as the rationale for assessing cardiopulmonary function as an outcome in this study?

- The authors need to be clear on whether they are assessing anxiety disorders / symptoms for the outcomes.

- Search terms from other anxiety related reviews and traditional chinese medicine should be employed in this review and cited as appropriate

-> https://pubmed.ncbi.nlm.nih.gov/33516963/

-> https://www.ncbi.nlm.nih.gov/pmc/articles/PMC4951626/

-> https://www.cureus.com/articles/127502-role-of-alternative-medical-systems-in-adult-chronic-kidney-disease-patients-a-systematic-review-of-literature

-> https://pubmed.ncbi.nlm.nih.gov/35104758/

Discussion

Potential limitations of this review should be included

-> e.g. language, limits of evaluating only RCTs

Language

- Significant amount of grammatical errors

-> To consider English editing services as appropriate

Reviewer #2: This paper describes the efficacy and safety of non-pharmacological therapy under the guidance of traditional Chinese medicine (TCM) theory in the treatment of anxiety in patients with myocardial infarction (MI) and provides a protocol for systematic review and meta-analysis. Authors assert that an aging population and increased work and life pressure is related to grater incidence of myocardial infarction and anxiety. Resultantly, patients’ quality of life is adversely impacted and risk of adverse cardiovascular events in patients with MI increases. Authors further contend that non- pharmacological therapy guided by traditional Chinese medicine (TCM) theory has been extensively used in China for the treatment anxiety and of coronary heart disease in community health service centers and tertiary hospitals. However, there is a need more research on TCM non-pharmacological therapies since current studies on such therapies are mainly single-center and small sample studies. The authors aim to address this gap in the literature with their study by synthesizing and further analyzing results of studies on the efficacy and safety of TCM non-pharmacological therapies for the treatment of

anxiety in patients with myocardial infarction. The review is as follows:

1. Authors should define ischemic heart disease for the lay reader and to keep consistency with the fact that the term anxiety is defined within the first paragraph.

2. Define psychotherapy and psychotropic drugs to provide context for the reader.

3. For the acronym ‘SSRIs’, write out the acronym in full when it is first introduced.

4. For discussion of traditional Chinese medicine, define ‘Taiji’ for the lay reader.

5. For type of interventions for the control group, the use of the word ‘sham acupuncture’ is curious. Authors should explain this.

6. For the search strategy, in “The two evaluators will independently conduct systematic searches on PubMed, Embase, Ovid, Scopus, Cochrane Central Controlled Trial Registration Center, Web of Science, China Knowledge Resource Integration Database and Wanfang Database from their establishment to December 31, 2022”, is December 31, 2022, the end date by which studies have to be published to be eligible? Is there a beginning date for eligibility?

7. Capitalize pronouns in ‘2.10 ethics and dissemination’, ‘2.11 patient and public involvement’ and ‘2.12 amendments (amending)’.

8. Discuss anticipated implications of your research.

Overall, this is a relevant topic that can make a unique contribution to the literature. Consider expanding discussion on the background and significance of the study, clearly detailing the steps in the research methodology and describing anticipated implications of this research.

7. PLOS authors have the option to publish the peer review history of their article (what does this mean?). If published, this will include your full peer review and any attached files.

Reviewer #1: No

Reviewer #2: No

---

## [Author Response · Author response to Decision Letter 0]

27 Mar 2023

Dear editor,

The comments of reviewers were extremely insightful and enabled us to greatly improve the quality of our manuscript. In the following pages are our point-by-point responses to each of the comments of the reviewers.We revised the manuscript with red color and made thorough revisions to the article in accordance with reviewers’ suggestions. 

We have made numerous corrections to the erroneous sentences and grammar in the original manuscript and have adjusted the article format and file name according to the style requirements of PLOS ONE. Since we have polished the language throughout the entire text, the overall structure of the sentences in “Revised Manuscript with Track Changes” has been adjusted. To make the content changes we made based on the reviewer's opinions more clear, we have marked pagexx and linexx for these modifications in the text following this reply (corresponding to the page and line in “Manuscript clean”). 

We hope that with these revisions and our accompanying responses, our manuscript is now suitable for publication in PLOS ONE.

We shall look forward to hearing from you at your earliest convenience.

Yours sincerely,

Weizhe Zhao 

E-mail: zhaoweizhe@126.com

To Additional Editor

1. The manuscript requires extensive edits for language, grammar and writing style.

Response：Thank you very much for your editing suggestions. We have made extensive revisions and refinements to the language, grammar, and writing style.

 2. "... biggest killer worldwide" - please use more formal and scientific language.

Response：I apologize for the grammatical and lexical errors in my previous statement. Here is the revised sentence: " According to an epidemiological report released by the World Health Organization (WHO) in December 2020, ischemic heart disease remains the leading cause of death globally, accounting for 16% of total deaths worldwide" Thank you for your suggestion. In page 3, line 51.

 3. The authors seem to use CHD and myocardial infarction interchangeably, please know that the two are not the same thing, one is a chronic disease and the other is an acute event.

Response：I'm sorry for some ambiguity in the words and content I wrote before. We have now unified the research object as patients with myocardial infarction.

 4. In the introduction, you should provide some reference to previous works and statistics on the prevalence of anxiety after cardiac arrest (citation: pubmed.ncbi.nlm.nih.gov/34826580).

Response：Thank you for your suggestion. We have now included statistical data on the prevalence of anxiety in patients with myocardial infarction in the article:

Studies have shown that the probability of depression and anxiety after myocardial infarction is about three times that of the general population. Over 50% of patients experience anxiety symptoms within a week after acute myocardial infarction. In page 3, line 56-58.

Data sources：

Lissåker CT, Norlund F, Wallert J, Held C, Olsson EM. Persistent emotional distress after a first-time myocardial infarction and its association to late cardiovascular and non-cardiovascular mortality. Eur J Prev Cardiol. 2019 Sep;26(14):1510-1518.

Wang J, Li P, Qin T, Sun D, Zhao X, Zhang B. Protective effect of epigallocatechin-3-gallate against neuroinflammation and anxiety-like behavior in a rat model of myocardial infarction. Brain Behav. 2020 Jun;10(6):e01633. doi: 10.1002/brb3.1633. Epub 2020 Apr 18.

 5. Were search terms in Mandarin used for the Chinese databases? Please specify.

Response：Thank you for your question, the search terms we used when searching the Chinese databases were in Chinese, and the detailed search terms have been elaborated in the search strategy (the search terms can be found in Supporting Information S2_ Search strategy)

6. How would the studies using different psychiatric scales e.g. HADS-A, HAMA, SAS, GAD-7, and DASS-A, be pooled?

Response：Thank you very much for your comments. For the different psychiatric rating scales, we will first conduct tests of heterogeneity, if I2 was greater than 50%, indicating the presence of statistical heterogeneity, a subgroup analysis or sensitivity analysis was conducted to eliminate the heterogeneity based on potential heterogeneity factors. If statistical heterogeneity persisted, but clinical homogeneity existed, a random-effects model was employed for conducting meta-analyses. In parallel, we also performed subgroup analyses for different psychiatric scales and applied sensitivity analyses to examine outcome stability. In page 9-10, “Data Analysis”, “Subgroup analysis”.

 7 "Concomitant with the rising social work pressure in people and the accelerated formation of an aging society" - I am not sure what is this supposed to mean. Please rephrase. 

Response：Sorry for these miswords and grammatical errors, the word description has now been revised. “With the increasing pressures of modern life and work, combined with a growing elderly population, the incidence of comorbid anxiety and myocardial infarction (MI) is on the rise.” In page 12, line 240-242.

 8. What exactly is "TCM theory"? This is mentioned a few times but authors did not go into any detail about TCM theory. 

Response：Thank you very much for your suggestion that the elucidation of TCM theory was indeed missing in previous articles now we have included in the article a detailed introduction of TCM theory:

Traditional Chinese Medicine (TCM) provides an explanation for the intricate relationship between myocardial infarction (MI) and emotional disorders from the perspective of Qi, blood, and meridians. TCM theory posits that the unobstructed flow of Qi and blood in the meridians is the foundation of life activities. Pathological conditions such as "blood stasis" and "Qi stagnation" obstructing the meridians can lead to various physical and mental illnesses. Among them, "blood stasis" is the primary pathological product of MI, which can cause obstruction of the heart meridian resulting in symptoms of chest pain and tightness, and can also affect the normal flow of Qi in the meridians, leading to abnormal emotions such as anxiety. To address this complex situation, TCM employs non-pharmacological therapies that differ from herbal treatment. These therapies can enhance psychological and emotional well-being by "regulating Qi and blood circulation in the meridians" while minimizing adverse reactions. In page 4-5, line 85-95.

Reviewer #1:

1. The title is slightly confusing and it is unclear if it is related to non pharmacological therapies or TCM.

It will benefit from amendment to reflect non-pharmacological TCM usage.

Response：I apologize for the lack of clarity in my previous explanation. In fact, traditional Chinese medicine treatment includes two categories: Chinese herbal medicine treatment and non-pharmaceutical physical therapy. The article mainly focuses on the latter. The concept and detailed introduction of non-pharmaceutical therapy in traditional Chinese medicine have been added to the article. In page 5, line 96-104.

Abstract

2. It may be better to change increase in people's life to "increase in people's life expectancy"

Response：Thanks for your help. We have deleted and corrected inappropriate sentences and words in the article.

3. The background section is confusing as it makes its less clear if the authors are inspecting patients with myocardial infarction or CHD

Response：Thank you very much for your valuable comments and constructive suggestions. Sorry, the description of included patients in the article may not be appropriate. The words used to describe included patients in the full text are now unified as patients with myocardial infarction.

4. Abstract - A brief sentence on why drug therapy for myocardial infarction and anxiety is controversial.

Response：Thanks for your help. Defects in drug treatment have been included in the abstract. As follows: The clinical use of selective serotonin reuptake inhibitors (SSRIs), which are commonly employed, in combination with antiplatelet drugs such as aspirin and clopidogrel, may increase the risk of bleeding in these patients. In page 1-2, line 21-23.

5. Abstract - The authors may wish to elaborate if they are focusing on anxiety symptoms or anxiety disorders or BOTH.

Response：Thank you very much for the valuable comments and constructive suggestion. This article focuses on anxiety symptoms. Relevant narratives have been sorted out in the full text to reduce misunderstandings.

6. Abstract - "Utilitarians" may be better worded as "practitioners"

Response：We apologize for such inappropriate words errors and the language of the full text has been polished.

7. Abstract - The methods does not describe any inclusion or exclusion criteria for this study in particular the PICO.

Response：Thank you for your great suggestion. Your comments are great importance to our article. The PICO of this study has been added to the abstract. As follows: To be eligible for inclusion, studies must diagnose patients with both myocardial infarction and anxiety, and patients must have received non-pharmacological Traditional Chinese Medicine (TCM) therapies such as acupuncture, massage, or Qigong, while the control group received standard treatments. The primary outcome measure will be changes in anxiety scores assessed by anxiety scales. In page2, line 34-38.

Introduction

8. I think it is important to discern if the authors are looking at as they are managed and treated differently.

Response：All the included patients will receive non-pharmacological under the guidance of TCM theory intervention. Their specific treatment may be different from acupuncture and moxibustion and massage, but they all belong to the category of non-pharmacological under the guidance of TCM theory intervention. We will use subgroup analysis to explore the source of heterogeneity for their specific treatment.

9. Likewise, discerning whether you are evaluating coronary heart disease patients / patients with recent myocardial infarction is important-> This is due to the title only reflecting patients with myocardial infarction.

Response：Thank you very much for your valuable comments and constructive suggestions. Sorry, the description of included patients in the article may not be appropriate. The words used to describe included patients in the full text are now unified as patients with myocardial infarction.

10. "In China, a considerable number of people have a certain resistance to psychotropic drugs, and they are more willing to choose some treatment methods with fewer side effects and higher safety" requires a citation

Response：Thank you for your suggestion. We have added the relevant references to the article accordingly. In page4, line 79.

11. The authors describe increased bleeding risk with SSRIs. However, what is the risk and has any recommendation been made by guidelines that SSRI should not be used?

- Clarification needs to be made as to which drugs are controversial to heart disease and its implications

Response：It is true that the guidelines do not explicitly state that SSRIs should not be used, but the application of SSRIs to patients with myocardial infarction and anxiety symptoms does have certain limitations. A specific description of bleeding risks has been added to the text. As follows: However, studies have revealed that selective serotonin reuptake inhibitors (SSRIs), a representative class of psychotropic medications, such as sertraline, fluoxetine, and fluvoxamine, carry a heightened risk of bleeding when co-administered with the most frequently used antiplatelet agents following a heart attack, including aspirin or clopidogrel, which can cause bleeding events such as gastrointestinal bleeding and hemorrhagic stroke. In page 4, line 71-76.

12. The authors write initially that the effect of psychotherapy on patients with anxiety and MI is unclear. They latter quote it is not satisfactory.-> There appears to be some contradictions here.

Response：Thank you very much for the valuable comments and constructive suggestion. We apologize for such inappropriate words errors and the language of the full text has been polished. As follows : However, studies have revealed that the current psychological therapies have limited efficacy in patients with heart disease and comorbid anxiety. In page 4, line 80-81.

13. Some definition needs to be made with regards to what TCM theory means and what non-pharmacological therapy the authors are looking at?

Response：The understanding of traditional Chinese medicine theory on myocardial infarction and anxiety and the concept of non-pharmacological therapy under the guidance of TCM theory will be explained in detail. In page 5, line 96-104.

14. The last paragraph in the introduction lacks citations

Response：References have been added to the last paragraph of the introduction. Such as: 

Liu H, He Y, Wang J, Miao J, Zheng H, Zeng Q. Epidemiology of depression at Traditional Chinese Medicine Hospital in Shanghai, China. Compr Psychiatry. 2016 Feb;65:1-8. doi: 10.1016/j.comppsych.2015.10.004

Amorim D, Amado J, Brito I, Fiuza SM, Amorim N, Costeira C, Machado J. Acupuncture and electroacupuncture for anxiety disorders: A systematic review of the clinical research. Complement Ther Clin Pract. 2018 May;31:31-37. doi: 10.1016/j.ctcp.2018.01.008

15. Given the heterogeneity of data collected, I am uncertain if a meta-analysis may be most appropriate for TCM trials

-> Perhaps a scoping review may be more appropriate.

Response：Meta-analysis can be applied to Traditional Chinese Medicine (TCM) trials, as it is a systematic method for integrating and analyzing the results of multiple independent studies. TCM trials can be conducted using rigorous scientific methods for experimental design, data collection, and statistical analysis, thus producing comparable and statistically meaningful results. Consequently, meta-analysis can be used to integrate and statistically analyze the results of multiple TCM trials, yielding more reliable and statistically significant conclusions.

We will screen and assess each study, excluding those of poor quality or non-standardized protocols, while ensuring the comparability of the data sources and study populations.

For example： 

Zhang J, Zhang Z, Huang S, Qiu X, Lao L, Huang Y, Zhang ZJ. Acupuncture for cancer-related insomnia: A systematic review and meta-analysis. Phytomedicine. 2022 Jul 20;102:154160.

Ji W, Wu L, Pan G, Zou X. Effects and Safety of Non-Pharmacological Therapies of Traditional Chinese Medicine for Coronary Heart Disease: An Overview of Systematic Reviews. Evid Based Complement Alternat Med. 2022 Mar 19;2022:8465269.

Methods

16. Was the PRISMA 2020 guidelines used?

Response：We apologize for the omission in the article. This meta-analysis will be carried out under the guidance of PRISMA 2020 guidelines, relevant content has been added to the article. As follow: It will adhere to the Cochrane Handbook and the Preferred Reporting Items for Systematic Reviews and Meta-analysis Protocols (PRISMA-P) guidelines. In page 7, line 148-149.

17. For the anxiety, were a psychiatrist involved in the diagnosis?

-> Noted that the authors mentioned a cut off of >=7 on HADS-A for diagnosis for anxiety

-> However, HADS-A is not used for diagnosis of anxiety disorders and instead valiated for diagnosis of anxity symptoms

Response：Thank you very much for the valuable comments and constructive suggestion. The main focus of the study was the symptoms of anxiety. So all patients included in this study have a clinical compliance anxiety scale score of anxiety level (such as HADS-A ≥ 7). The reason we included patients with anxiety disorders is because these patients all have anxiety symptoms. To avoid missing any relevant information, we incorporated studies focusing on patients with anxiety disorders and analyzed the changes in anxiety scales found within their articles. The diagnoses of anxiety disorders in the studies that have been included are explicitly determined by psychiatrists who are part of the research process, rather than by us.

18. What are the exclusion for participants in the study?

Response：We apologize for the omission in the article. We have now improved the exclusion criteria in the study participants section of the text. As follow: Exclusion criteria: (1) patients with severe heart failure who are unable to receive the research intervention. (2) patients with a history of mania, consciousness disorders, or other serious mental illnesses. (3) literature that cannot provide detailed outcome indicators. (4) duplicate publications. In page 7 line 130-132.

19. What as the rationale for assessing cardiopulmonary function as an outcome in this study?

Response：Assessing cardiopulmonary function can provide a better understanding of the cardiac function status of patients. It is used as one of the outcome measures in the study to investigate the effects of non-pharmacological Traditional Chinese Medicine therapies on cardiac function in patients with acute myocardial infarction and comorbid anxiety.

20. The authors need to be clear on whether they are assessing anxiety disorders / symptoms for the outcomes.

Response：Thank you very much for the valuable comments and constructive suggestion. The main focus of the study was the symptoms of anxiety. So all patients included in this study have a clinical compliance anxiety scale score of anxiety level (such as HADS-A ≥ 7), and we will use the changes in anxiety scale scores as the primary outcome measure.

21. Search terms from other anxiety related reviews and traditional chinese medicine should be employed in this review and cited as appropriate

-> https://pubmed.ncbi.nlm.nih.gov/33516963/

-> https://www.ncbi.nlm.nih.gov/pmc/articles/PMC4951626/

-> https://www.cureus.com/articles/127502-role-of-alternative-medical-systems-in-adult-chronic-kidney-disease-patients-a-systematic-review-of-literature

-> https://pubmed.ncbi.nlm.nih.gov/35104758/

Response：We greatly appreciate your suggestion. We have already added relevant terms to the main search keywords section. Detailed and specific anxiety-related search terms and Traditional Chinese Medicine non-pharmacological therapy search terms are listed in the search strategy appendix. Additionally, we have cited reference： https://pubmed.ncbi.nlm.nih.gov/35104758

22. Potential limitations of this review should be included

-> e.g. language, limits of evaluating only RCTs

Response：Thank you for your suggestion. We have incorporated potential limitations, such as language and types of included literature, into the discussion section:

This study has some potential limitations. In terms of language, we only included studies in Chinese and English, and other studies using non-pharmacological TCM therapies in different languages could not be effectively incorporated. Additionally, this study only evaluated the results of RCTs and did not conduct a comprehensive analysis of other types of studies such as cohort studies and observational studies. This study lays a foundation for future research and provides new evidence-based medicine support for clinical application. Future studies could expand the breadth of research, include studies in different languages and types, fully consider limitations and biases, and conduct more in-depth and reasonable research and discussion. In page12-13.

Language

23. Significant amount of grammatical errors

-> To consider English editing services as appropriate

Response：We apologize for the presence of these grammatical errors. We have implemented extensive revisions to correct the grammar and refine the language.

Reviewer #2:

1. Authors should define ischemic heart disease for the lay reader and to keep consistency with the fact that the term anxiety is defined within the first paragraph.

Response：I'm sorry for the confusion caused by the ambiguous wording and content in our previous writing. We have now clarified that the research will focus on patients with myocardial infarction.

2.Define psychotherapy and psychotropic drugs to provide context for the reader.

Response：Thank you very much for your suggestion. We have now added relevant descriptions of psychological therapies and psychotropic medications used for treating anxiety in the article. In page4, line 68-71.

3. For the acronym ‘SSRIs’, write out the acronym in full when it is first introduced.

Response：I'm sorry for the mistake. The necessary changes have been made in the article.

4. For discussion of traditional Chinese medicine, define ‘Taiji’ for the lay reader.

Response：Thank you very much for your suggestion. We have now provided an explanation of the concept and characteristics of non-pharmaceutical therapies in traditional Chinese medicine, such as Tai Chi and other exercises. In page5, line 96-104.

5. For type of interventions for the control group, the use of the word ‘sham acupuncture’ is curious. Authors should explain this.

Response：Thank you very much for the question, "Sham acupuncture" is a commonly used control group in acupuncture clinical trials. It involves the use of a non-penetrating needle, also known as a placebo needle or a blunt needle, that is designed to mimic the sensation of real acupuncture without actually penetrating the skin or stimulating the acupuncture points. The purpose of using a sham acupuncture control group is to help researchers determine the true efficacy of real acupuncture by minimizing the placebo effect and any potential biases.

6. For the search strategy, in “The two evaluators will independently conduct systematic searches on PubMed, Embase, Ovid, Scopus, Cochrane Central Controlled Trial Registration Center, Web of Science, China Knowledge Resource Integration Database and Wanfang Database from their establishment to December 31, 2022”, is December 31, 2022, the end date by which studies have to be published to be eligible? Is there a beginning date for eligibility?

Response：I'm sorry for the unclear language in my previous description. The research included articles published up to December 31, 2022. The sentence has been revised accordingly. In page7-8, line 149-153.

7. Capitalize pronouns in ‘2.10 ethics and dissemination’, ‘2.11 patient and public involvement’ and ‘2.12 amendments (amending)’.

Response：I'm sorry for the writing errors. The necessary corrections have been made in the text.

8. Discuss anticipated implications of your research.

Overall, this is a relevant topic that can make a unique contribution to the literature. Consider expanding discussion on the background and significance of the study, clearly detailing the steps in the research methodology and describing anticipated implications of this research.

Response：Thank you very much for your suggestion! We hope that this research will provide higher-level clinical evidence for the safety and effectiveness of non-pharmaceutical therapies in traditional Chinese medicine for treating anxiety in patients with myocardial infarction, and offer more treatment options for clinical doctors. We have further added a discussion of the background and significance of the research and provided a more detailed explanation of the research plan.

---

## [Decision Letter · Decision Letter 1]

19 Apr 2023

PONE-D-22-34677R1Efficacy and safety of non-pharmacological therapy under the guidance of TCM theory in the treatment of anxiety in patients with myocardial infarction: a protocol for systematic review and meta-analysisPLOS ONE

Dear Dr. Zhao,

Thank you for submitting your manuscript to PLOS ONE. After careful consideration, we feel that it has merit but does not fully meet PLOS ONE’s publication criteria as it currently stands. Therefore, we invite you to submit a revised version of the manuscript that addresses the points raised during the review process.

Some comments were not acted upon:1. close edits for language still necessary throughout the manuscript2. in the introduction, you should provide some reference to previous works and statistics on the prevalence of anxiety after cardiac arrest (citation: pubmed.ncbi.nlm.nih.gov/34826580). Ref [5] is not specific for anxiety and a meta-analysis would provide a better estimate than a single study.3. how would the studies using different psychiatric scales e.g. HADS-A, HAMA, SAS, GAD-7, and DASS-A, be pooled?4. how does the authors intend to handle the high study heterogeneity in the meta-analysis? Would provisions be made for a systematic review without meta-analysis?

We look forward to receiving your revised manuscript.

Kind regards,

Qin Xiang Ng, MD, MPH

Academic Editor

PLOS ONE

Reviewers' comments:

Reviewer's Responses to Questions

**Comments to the Author**

1. Does the manuscript provide a valid rationale for the proposed study, with clearly identified and justified research questions?

Reviewer #1: Yes

Reviewer #2: Yes

2. Is the protocol technically sound and planned in a manner that will lead to a meaningful outcome and allow testing the stated hypotheses?

Reviewer #1: Partly

Reviewer #2: Yes

3. Is the methodology feasible and described in sufficient detail to allow the work to be replicable?

Reviewer #1: Yes

Reviewer #2: Yes

4. Have the authors described where all data underlying the findings will be made available when the study is complete?

Reviewer #1: Yes

Reviewer #2: Yes

5. Is the manuscript presented in an intelligible fashion and written in standard English?

Reviewer #1: No

Reviewer #2: Yes

6. Review Comments to the Author

You may also provide optional suggestions and comments to authors that they might find helpful in planning their study.

Reviewer #1: Dear editor,

Thank you for the kind invitation to review the following mansucript.

The authors have made considerable efforts to amend their manuscript.

Some important methodological concerns remains about the manuscript.

Point 15 of Reviewer 1

-> The authors informed that meta-analysis can be performed due to it being used for other studies

-> It is generally well accepted that significant heterogenity exists for TCM related trials

 How do the authors plan to account for the methodological, clinical heterogenity of included studies?

 The authors need to understand that meta-analyses if done inappropriately, will yield non-useful clinical information

Point 17: The authors informed that they included HADS-A to screen for anxiety symptoms among patients with anxiety disorders

-> While anxiety symptoms can be screened by HADS-A, it is more used as a screening tool for anxiety symptoms in the general population than patients with anxiety. Further clarification needs to be made.

-> If HADS-A is used, what is defined as the Minimally clincially important difference?

-> How is the cutoff of 7 decided and not 8 used for other studies?

Point 18: Clarification needs to be made about what consciousness disorders and other serious mental illness entails.

Point 21: Please make the necessary citation as they contain important search terms relevant to your article and will add to the robustness of your review

-> https://pubmed.ncbi.nlm.nih.gov/33516963/

-> https://www.ncbi.nlm.nih.gov/pmc/articles/PMC4951626/

-> https://pubmed.ncbi.nlm.nih.gov/28064110/

-> https://www.cureus.com/articles/127502-role-of-alternative-medical-systems-inadult-chronic-kidney-disease-patients-a-systematic-review-of-literature

-> https://pubmed.ncbi.nlm.nih.gov/35104758/

Significant Grammatical errors exist currently in the manuscript. It will benefit from English Editing Service before it can be published

Reviewer #2: The revised manuscript is clearer and more detailed and appears suitable for publication. The authors have expanded on their discussion and provided definitions and explanations on terms that may be helpful for the lay reader.

7. PLOS authors have the option to publish the peer review history of their article (what does this mean?). If published, this will include your full peer review and any attached files.

Reviewer #1: No

Reviewer #2: No

---

## [Author Response · Author response to Decision Letter 1]

19 May 2023

To Reviewer 1

Dear Reviewer,

Thank you for taking the time to review our manuscript and providing your constructive feedback. We sincerely appreciate your valuable comments and suggestions. In accordance with your feedback, we have revised the manuscript and subjected the language to rigorous editing and refinement through a professional English editing service (S3_Certificate of editing). The specific modifications to the content of the article are as follows:

1.-> It is generally well accepted that significant heterogenity exists for TCM related trials

 How do the authors plan to account for the methodological, clinical heterogenity of included studies? 

Response：Thank you for your critical questions. We acknowledge that trials related to traditional Chinese medicine often exhibit significant heterogeneity, which is a factor we must consider in conducting a meta-analysis. To address the methodological and clinical heterogeneity in the included studies, we propose the following measures:1.Methodological heterogeneity resolution: In selecting studies, we will adhere to rigorous quality assessment criteria, incorporating only randomized controlled trials (RCTs) to ensure consistency in our research design and implementation. Furthermore, we will use risk of bias tools from networks such as the Cochrane Collaboration to evaluate the quality of each study, thereby controlling the potential impact of study quality. 2.Clinical heterogeneity resolution: We will standardize the criteria for including studies and strictly adhere to these criteria. During the data collection phase, we will meticulously record various clinical information, such as the treatment methods received by patients and the employed assessment scales. While the specific interventions that patients receive may vary, all these therapeutic methods are rooted in the theories of Qi and blood, along with the meridian system as articulated in Traditional Chinese Medicine. Hence, they share a unifying core treatment philosophy. Beyond this, we will employ subgroup analysis and meta-regression to explore and explain these clinical heterogeneities. For instance, we will conduct further subgroup analyses based on different types of treatment methods. Besides, we will conduct a sensitivity analysis to test the robustness of our results. If the removal of a particular study leads to significant changes in the outcome, we will delve into a thorough discussion of the implications. From page10, line 199 to page12, line 243.( Data Analysis; Subgroup analysis; Sensitivity Analysis; Grading the Quality of Evidence)

2.1The authors informed that they included HADS-A to screen for anxiety symptoms among patients with anxiety disorders

-> While anxiety symptoms can be screened by HADS-A, it is more used as a screening tool for anxiety symptoms in the general population than patients with anxiety. Further clarification needs to be made.

-> If HADS-A is used, what is defined as the Minimally clincially important difference?

Response：Thank you for highlighting the need for further clarification regarding the use of HADS-A to screen for anxiety symptoms in our study. We appreciate your expertise in this area and acknowledge that HADS-A is more commonly used as a screening tool for anxiety symptoms in the general population. The purpose of listing HADS-A in the manuscript was to provide an example for " Patients with a clinical compliance anxiety scale score of anxiety level " rather than to evaluate all patients with anxiety symptoms. As some of the included studies involve patients with anxiety disorders, they often employ the HAMA scale for assessment. To avoid ambiguity, we have further clarified and revised this part of the manuscript. It is worth noting that our research primarily focuses on the changes in anxiety symptoms; hence, we included patients with anxiety disorders, as they exhibit these symptoms. To avoid omitting any relevant information, we incorporated studies involving myocardial infarction patients with concurrent anxiety disorders and analyzed changes in anxiety scales reported in their articles.(The diagnoses of anxiety disorders in the included studies were determined by psychiatrists involved in the research process) For evaluating different scales, we use the SMD to account for the variability between them by standardizing the results, enabling us to integrate and compare their outcomes. In page6, line 124-128.

2.2-> If HADS-A is used, what is defined as the Minimally clincially important difference?

-> How is the cutoff of 7 decided and not 8 used for other studies?

Response：We sincerely apologize for the oversight in our manuscript. It was an unintended error that we mistakenly wrote HADS-A ≥ 7 instead of HADS-A > 7. We understand that this may have caused confusion in your interpretation of our findings. We have now corrected this error in the manuscript to accurately reflect "HADS-A > 7." In page6, line 125. Thank you for your understanding.

3.Clarification needs to be made about what consciousness disorders and other serious mental illness entails.

Response：Thank you for your suggestion. We have further clarified the consequences of altered consciousness and other severe mental illnesses in the manuscript: Patients with a history of mania, consciousness disorders, or other serious mental illnesses include schizophrenia, bipolar disorder, and severe major depressive disorder. Patients with consciousness disorders may be unable to accurately communicate for scale evaluations, and other severe mental illnesses could potentially influence participants' responses to treatment or interfere with anxiety symptoms, thereby affecting research outcomes. Furthermore, the efficacy of traditional Chinese medicine on this patient population remains unclear. Excluding them helps mitigate the risk of exacerbating their conditions. In page7, line 132-137.

4.Please make the necessary citation as they contain important search terms relevant to your article and will add to the robustness of your review

-> https://pubmed.ncbi.nlm.nih.gov/33516963/

-> https://www.ncbi.nlm.nih.gov/pmc/articles/PMC4951626/

-> https://pubmed.ncbi.nlm.nih.gov/28064110/

-> https://www.cureus.com/articles/127502-role-of-alternative-medical-systems-inadult-chronic-kidney-disease-patients-a-systematic-review-of-literature

-> https://pubmed.ncbi.nlm.nih.gov/35104758/

Response：Thank you for your valuable suggestions and guidance, including the recommendation to incorporate additional references. We have carefully reviewed the suggested articles and have included the following references in our manuscript to enhance the robustness of our review.

https://pubmed.ncbi.nlm.nih.gov/33516963/

https://www.ncbi.nlm.nih.gov/pmc/articles/PMC4951626/

https://www.cureus.com/articles/127502-role-of-alternative-medical-systems-inadult-chronic-kidney-disease-patients-a-systematic-review-of-literature

(In page8, line 161-162, as well as Table 1.and other databases in the “S_2 Search strategy”)

https://pubmed.ncbi.nlm.nih.gov/35104758/ (In page13, line 275-276.)

Once again, thank you for your invaluable feedback.

To Reviewer 2

Dear Reviewer,

We greatly appreciate your positive feedback on the revised manuscript and are pleased to know that our efforts to improve clarity and provide more detailed explanations have met your expectations. Now we have meticulously revised and polished the language of the manuscript through a professional English editing service. We are grateful for your guidance and support throughout the revision process, which has contributed to enhancing the quality of our work. Thank you for considering our manuscript suitable for publication.

---

## [Decision Letter · Decision Letter 2]

14 Jun 2023

PONE-D-22-34677R2

Efficacy and safety of non-pharmacological therapy under the guidance of TCM theory in the treatment of anxiety in patients with myocardial infarction: a protocol for systematic review and meta-analysis

PLOS ONE

Dear Dr. Zhao,

Thank you for submitting your manuscript to PLOS ONE. After careful consideration, we feel that it has merit but does not fully meet PLOS ONE’s publication criteria as it currently stands. Therefore, we invite you to submit a revised version of the manuscript that addresses the points raised during the review process.

Please address the journal requirements found below. 

We look forward to receiving your revised manuscript.

Kind regards,

Qin Xiang Ng, MD, MPH

Academic Editor

PLOS ONE

Journal Requirements:

Because of the varying quality of evidence supporting TCM pathologies and therapies, care should be taken to avoid statements indicating their effectiveness or that encouraging the use of TCM diagnoses in medical care. Theories of TCM can be discussed provided they are framed as such. However, the manuscript should not include statements indicating that TCM therapies are effective. The manuscript should not include presentation of TCM theories presented as fact. Specific examples, from lines 86-103 that require revision can be found below:

1) "Pathological conditions such as "blood stasis" and "Qi stagnation" obstructing the meridians can lead to various physical and mental illnesses" - should be revised to "TCM theory suggests that pathological conditions such as "blood stasis" and "Qi stagnation" obstructing the meridians can lead to various physical and mental illnesses" or similar. A suitable reference must be provided.

2) A similar edit is required to the sentence ""Blood stasis" is the primary pathological product of MI, which can obstruct the heart meridian resulting in symptoms of chest pain and tightness, it can also affect the normal flow of Qi in the meridians, leading to abnormal emotions such as anxiety". A reference must be required.

3) The following sentence is missing a reference. "These therapies can enhance psychological and emotional well-being by "regulating Qi and blood circulation in the meridians" while minimizing adverse reactions." A reference must be provided and the text updated to indicate that "Evidence has suggested that these therapies..." or similar.

4) Several other statements from lines 86-103 are missing references. Reference 16 does not appear to support the claim made. Please carefully revise the text in this section to provide suitable references for all statements made

Finally, Please review your reference list to ensure that it is complete and correct. If you have cited papers that have been retracted, please include the rationale for doing so in the manuscript text, or remove these references and replace them with relevant current references. Any changes to the reference list should be mentioned in the rebuttal letter that accompanies your revised manuscript. If you need to cite a retracted article, indicate the article’s retracted status in the References list and also include a citation and full reference for the retraction notice.

Reviewers' comments:

Reviewer's Responses to Questions

**Comments to the Author**

1. Does the manuscript provide a valid rationale for the proposed study, with clearly identified and justified research questions?

Reviewer #1: Yes

2. Is the protocol technically sound and planned in a manner that will lead to a meaningful outcome and allow testing the stated hypotheses?

Reviewer #1: Yes

3. Is the methodology feasible and described in sufficient detail to allow the work to be replicable?

Reviewer #1: Yes

4. Have the authors described where all data underlying the findings will be made available when the study is complete?

Reviewer #1: Yes

5. Is the manuscript presented in an intelligible fashion and written in standard English?

Reviewer #1: Yes

6. Review Comments to the Author

You may also provide optional suggestions and comments to authors that they might find helpful in planning their study.

Reviewer #1: I have no further comments. The changes are satisfactory and manuscript suitable for publication .

7. PLOS authors have the option to publish the peer review history of their article (what does this mean?). If published, this will include your full peer review and any attached files.

Reviewer #1: No

---

## [Author Response · Author response to Decision Letter 2]

17 Jun 2023

Dear Reviewer,

We greatly appreciate your positive feedback on the revised manuscript and are pleased to know that our efforts to improve clarity and provide more detailed explanations have met your expectations. We are grateful for your guidance and support throughout the revision process, which has contributed to enhancing the quality of our work. Thank you for considering our manuscript suitable for publication.

---

## [Editor Report · Decision Letter 3]

21 Jun 2023

Efficacy and safety of non-pharmacological therapy under the guidance of TCM theory in the treatment of anxiety in patients with myocardial infarction: a protocol for systematic review and meta-analysis

PONE-D-22-34677R3

Dear Dr. Zhao,

We’re pleased to inform you that your manuscript has been judged scientifically suitable for publication and will be formally accepted for publication once it meets all outstanding technical requirements.

Kind regards,

Qin Xiang Ng, MD, MPH

Academic Editor

PLOS ONE
---

## [Editor Report · Acceptance letter]

26 Jun 2023

PONE-D-22-34677R3 

Efficacy and safety of non-pharmacological therapy under the guidance of TCM theory in the treatment of anxiety in patients with myocardial infarction: a protocol for systematic review and meta-analysis 

Dear Dr. Zhao:

I'm pleased to inform you that your manuscript has been deemed suitable for publication in PLOS ONE. Congratulations! Your manuscript is now with our production department. 

Kind regards, 

on behalf of

Dr. Qin Xiang Ng 

Academic Editor

PLOS ONE